

# Gas chromatography-mass spectrometry and Fourier-transform infrared spectroscopy coupled to chemometrics for metabolome analysis of different milk types in the light of green analytical chemistry

Mohamed A. Farag[1], Sherif M. Eid[2] and Sherine El-Shamy[3]

[1] Pharmacognosy, Cairo University, Cairo, Egypt
[2] Analytical Chemistry/ Pharmacy, October 6 University, Giza, Egypt
[3] Pharmacognosy/Pharmacy, Modern University for Technology and Information, Cairo, Egypt

## ABSTRACT

**Background**. Milk is an important dietary source for a healthy and balanced diet, owing to its rich content of proteins, fats, sugars, vitamins, and minerals. Due to the importance of milk macronutrient content for consumers' preferences, a multiplex metabolomics-based approach using Fourier-transform infrared spectroscopy (FTIR) and gas chromatography-mass spectrometry (GC-MS) fingerprinting platforms was employed for the characterization of metabolites in different types of buffalo (BM), cow (CM), goat (GM), and camel (LM) milk.

**Methods**. GC-MS and FTIR coupled to chemometric tools (multivariate data analysis) were employed for the discriminative qualitative and quantitative analysis of BM, LM, CM, and GM milk, targeting their primary metabolites. A side-by-side comparative assessment of the performance of both the FTIR and GC-MS methods was implemented in the light of green analytical chemistry principle (GAC) and white analytical chemistry principles (WAC) using the 12 green analytical chemistry principles (12 GAC), analytical greenness metric approach (AGREE), national environmental methods index (NEMI), eco-scale assessment (ESA) and complementary green analytical procedure index (ComplexGAPI), and the red green blue (RGB) 12 algorithms.

**Results**. The milk types were qualitatively identified by visual inspection of their characteristic FTIR spectra as a fingerprint for each milk type. Quantitatively, GC-MS revealed the presence of 87 peaks belonging to alcohols, amino acids/nitrogenous compounds, fatty acids, organic acids, sterols, sugars, and vitamins. Sugars, mainly lactose, appeared as the major component in all milk types. The highest lactose content was detected in CM 1.07-fold higher than LM making LM a potential alternative for lactose intolerance. Both BM and CM were found to contain the highest organic acid content 5.2-fold higher than that in LM, accounting for their acidity (sourness), while the lowest level was found in LM. On the other hand, LM had the highest vitamins content compared to other milks. Lastly, FTIR outperformed GC-MS in terms of greenness and whiteness, suggesting its utilization as an alternative to traditional chromatographic techniques such as GC-MS.

Corresponding author
Mohamed A. Farag,
mohamed.farag@pharma.cu.edu.eg

# INTRODUCTION

Milk is considered an important dietary source for a healthy and balanced diet, owing to its richness in proteins, fats, sugars, vitamins, and minerals (*Farag et al., 2021*). Global milk production is dominated by cow milk (CM) (81%), followed by buffalo milk (BM) (15%) as reported by the Food and Agriculture Organization (FAO) and the Organization for Economic Cooperation (OECD), in addition to goat milk (GM) (2.3%) and camel milk (LM) (0.37%), which can be transformed into a variety of dairy products (*Konuspayeva, Faye & Duteurtre, 2022*; *Lund & Ahmad, 2021*; *Mejares, Huppertz & Chandrapala, 2022*). Milk exhibits a pleasant, slightly sweet taste, accompanied by a faint aroma and a pleasant aftertaste. Milk components and flavours are affected by several factors, such as the dairy animal's physiological condition, animal feed type, biological, and enzymatic changes in milk, in addition to the environment surrounding the milking area (*Al-Attabi, D'arcy & Deeth, 2008*).

In Western culture, CM is the most commonly used milk, described to have a mild flavor and creamy sweet note. On the other hand, BM is more commonly used in human nutrition in the developing countries such as India and Egypt (*El-Salam, Mohamed & El-Shibiny, 2011*). It was found that milk components such as fats (7–9), lactose (5), and proteins (3–5) (g/100 g milk) are more abundant in BM compared to CM (*Coolbear et al., 2022*). Furthermore, milk products such as mozzarella cheese and ghee are considered to be BM specialties (*El-Salam, Mohamed & El-Shibiny, 2011*). GM contains the least lactose content compared to both CM and BM, indicating its possible suitability for patients suffering from lactose intolerance (*Kapadiya et al., 2016*; *Meena, Rajput & Sharma, 2014*). However, due to its peculiar distinctive unpleasant odour (*Jia et al., 2020*), GM might discourage customers' preferences.

LM is regarded as a vital nutritional resource in the deserts of Asia and Africa, and it is increasingly recognized as a promising alternative for feeding both infants and adults, as well as for the development of various dairy-based products. Despite its nutritive value and several health benefits, it has not gained as much attention as CM (*Bakry et al., 2021*; *Ho, Zou & Bansal, 2022*). LM has a dark white colour, sweet odour, and a sharp salty taste that varies according to the drinking water abundance for camels as well as the feed type, as typical in most milk types (*Hammam, 2019*). Compared to CM, LM is characterized by a higher content of moisture, iron, vitamin C, and proteins, alongside a reduced lactose concentration (*Farag, El Hawary & Elmassry, 2020*). Furthermore, LM content of odd and branched chain fatty acids alongside the low ratios of polyunsaturated fatty acids (n-6 to n-3), potentiate LM to be considered a promising functional food (*Wang et al., 2022*). LM shows high fat digestibility compared with BM and CM (*Meena, Rajput & Sharma, 2014*). Moreover, LM has been associated with a range of health benefits in individuals affected by autoimmune diseases, tuberculosis, metabolic disorders, hepatitis, liver cirrhosis, cancer,

Crohn's disease, diabetes, autism, rickets, and rotavirus-induced diarrhea (*Hammam, 2019*).

Being a non-destructive simple technique, covering a wide spectral range (*Attia et al., 2023*), several infrared (IR) spectroscopic-based methods have been developed for the qualitative and quantitative analysis of milk (*Balan et al., 2020*; *Conceição et al., 2018*; *Eid, el Shamy & Farag, 2022*; *Gomes Marques De Freitas et al., 2021*; *Goulden, 2009*; *Pralle & White, 2020*). Fourier-transform infrared spectroscopy (FTIR) coupled to chemometric tools facilitated the quantitative determination of major milk components (*Bahadi, Ismail & Vasseur, 2021*; *Martel, Paquin & Bertrand, 2009*). FTIR was used for identifying adulterants in milk with high accuracy such as extraneous water, urea (*Saji et al., 2024*), foreign proteins (*Souhassou et al., 2018*), melamine, starch, vegetable or animal fats (*Poonia et al., 2016*). Subsequently, FTIR was adopted as the standard analytical approach for milk assessment using MilkoScan instruments, offering both quantitative and qualitative molecular fingerprints through the identification of characteristic functional groups (*Milk, 2000*; *Sánchez et al., 2007*).

Gas chromatography-mass spectrometry (GC-MS) is considered an essential metabolomics technique for the qualitative and quantitative analysis of primary metabolites, with increased sensitivity for detecting compounds at lower concentrations (*Baky et al., 2022*). Currently, GC–MS remains one of the most widely employed and effective analytical techniques in metabolomics, owing to its robustness, high separation efficiency, selectivity, and sensitivity (*Beale et al., 2018*). In addition, chemometric techniques were applied to enhance the interpretation of GC–MS metabolite datasets, facilitating the exploration of sample heterogeneity, marker identification, and classification. Among these, unsupervised and supervised multivariate data analysis methods—such as principal component analysis (PCA), hierarchical cluster analysis (HCA), and orthogonal partial least squares discriminant analysis (OPLS-DA)—are commonly utilized for effective data visualization and pattern recognition (*Fahmy, El-Shamy & Farag, 2023*).

Due to the importance of milk macronutrient content for consumers' preferences, in addition to milk flavor, FTIR and GC-MS coupled to multivariate data analysis were utilized for the qualitative and quantitative analysis of BM, CM, GM, and LM. The aim of this study was to profile primary metabolites in milk from different types as presented herein for the first time aided by multivariate data analysis for samples' classification.

Consequently, the present study has two primary objectives. Firstly, GC–MS and FTIR were introduced as complementary analytical techniques for the identification and quantification of milk constituents in BM, LM, CM, and GM samples, with the added capability of distinguishing among the different milk types. Secondly, the development of a comparative evaluation of the greenness and whiteness of the employed methods FTIR and GC-MS, utilizing the 12 principles of Green Analytical Chemistry (GAC), Analytical GREEnness metric approach (AGREE), Eco-Scale Assessment (ESA), National Environmental Methods Index (NEMI), Complementary Green Analytical Procedure Index (ComplexGAPI), and the Red Green Blue (RGB) twelve algorithms. The analytical methods (FTIR and GC-MS) greenness assessments aim to provide insights into their relative environmental impact and suitability for analytical applications by assessing their

safety, automation, cost, energy and solvent consumption, and wastes production. With the aim to introduce a cost-effective, sensitive, eco-friendly, and economical alternative to the traditional methods known for their non-green nature and high costs, particularly in the field of food analysis.

Furthermore, the developed FTIR method can be used as complementary to the developed GC-MS method to enhance quality control and ensure the authenticity of milk products, particularly in distinguishing poorly labeled consumer products, such as detecting mislabeled milk types. Additionally, the study provides insights that can support compliance with regulatory guidelines of food quality control and its safety. Lastly, it contributes to metabolites profiling, which could have potential implications for human or animal health. Evaluating the greenness of analytical methods is essential, as it aligns with the principles of sustainable development by reducing environmental impact and conserving resources. Greener methods ensure safer practices for both operators and the environment, while also reducing waste and costs, making them highly suitable for routine applications in food analysis, particularly in economically constrained regions.

## MATERIALS & METHODS

### Milk samples

Milk samples were obtained at the farm level, where individual farmers provided fresh milk collected from cows, buffaloes, goats, and camels across various farms. Prior to sampling, the milk was thoroughly stirred in its container for 10 min to ensure homogeneity. Aliquots were then drawn from the surface of the container using a sterilized dipper and transferred into sterile test tubes. The collected samples were immediately stored at −20 °C until further analysis. A total of 32 milk samples were collected in total, eight milk specimens from each milk type different sources (BM 1-8, CM 1-8, GM 1-8, and LM 1-8) were used in this study, their codes are listed in Table S1.

### Instruments

GC–MS analysis was carried out using a Shimadzu GC-17A gas chromatograph coupled with a Shimadzu QP5050A quadrupole mass spectrometer. Separation was achieved on an Rtx-5MS capillary column (30 m length, 0.25 mm internal diameter, 0.25 μm film thickness). Samples were injected in splitless mode for 30 s, with the injector temperature maintained at 220 °C. The column oven temperature was initially set at 38 °C. Helium was used as the carrier gas. Mass spectra were recorded using electron ionization (EI) at 70 eV over a mass range of m/z 35–500. FTIR device model IRAffinity-1 (Shimadzu, Kyoto, Japan) in the range of 4,600 to 400 cm$^{-1}$. The ATR Unit was purged using nitrogen gas to remove any interference from $CO_2$ or air humidity.

### GC-MS analysis of silylated primary metabolites

The analysis of the primary metabolites (alcohols, amino acids/ nitrogenous compounds, fatty acids, organic acids, sterols, sugars, sugar acids, sugar alcohol, and vitamins) was performed following the procedure cited in *Baky et al. (2022)*. 2.5 mL of each milk sample was extracted using five mL pure methanol to precipitate proteins and sonicated for 30

min with continuous vortex shaking. An aliquot of 100 µL of the methanolic extract was transferred into a screw-cap vial and evaporated to complete dryness under a gentle stream of nitrogen gas. For derivatization, the dried residue was reconstituted in 150 µL of N-methyl-N-(trimethylsilyl)trifluoroacetamide (MSTFA), previously diluted with anhydrous pyridine, and incubated at 60 °C for 45 min. Chromatographic separation was carried out using a Shimadzu GC-17A gas chromatograph coupled to a Shimadzu QP5050A quadrupole mass spectrometer. An Rtx-5MS capillary column (30 m length, 0.25, mm inner diameter, 0.25, µm film thickness) was employed for the analysis. Sample injections were performed in splitless mode with a purge time of 30 s. The injector temperature was set to 220 °C. The column oven was held at 40 °C for 3 min. The temperature was then increased at a rate of 12 °C/min to 180 °C, held for 5 min, and then increased at a rate of 40 °C/min to 220 °C, held for 2 min. Helium carrier gas flow rate was at one mL/min. Mass spectra were obtained by electron ionization at 70 eV, using a spectral range (35–500) m/z (*El-Shabasy et al., 2024*). Three different specimens from each milk sample were analysed under similar conditions for biological replicates assessment.

## Identification of primary metabolites and multivariate data analysis

Identification of primary silylated metabolites was performed by comparing their retention indices (RI), calculated relative to a series of co-injected n-alkanes ($C_8$–$C_{30}$) under identical chromatographic conditions. Compound identification was further supported by mass spectral matching against the WILEY and NIST library databases, and confirmed using authentic standards when available. Prior to spectral matching, peak deconvolution was conducted using the Automated Mass Spectral Deconvolution and Identification System (AMDIS; http://www.amdis.net/). Peak abundances (peak area of the total ion chromatogram) were expressed as fold differences. Peak abundance mass lists generated using the MS dial program version 4.0 (*Saied et al., 2023*) with GC-MS analysis software, were exported to multivariate data analysis using SIMCA-P software. All variables were mean-centered and scaled to Pareto variance before modelling. The unsupervised principal component analysis (PCA) was performed initially to provide a general overview of the variance among metabolites in milk specimens.

To further validate the findings from PCA and facilitate biomarker identification, supervised orthogonal partial least squares discriminant analysis (OPLS-DA) was applied. The performance of the chemometric models was assessed using the $R^2$ and $Q^2$ parameters, where $R^2$ indicates the goodness-of-fit and $Q^2$ reflects the predictive ability of the model. Outlier detection was carried out using the distance to the model in X-space (DModX). Additionally, an iterative permutation test was performed to evaluate the statistical significance of group separation and to rule out the possibility of random discrimination (*Baky et al., 2022*).

## Water activity and moisture content of milk samples

The water activity ($a_w$) of milk samples was determined at 25 °C using Aqualab 4TE Aqualab, Pullman, CA, USA). The moisture content (MC) of the milk samples was determined gravimetrically. The samples were transferred to an air oven (Thermo Electron

Corporation, Waltham, MA, USA) and dried at 100 °C to constant weight for 2 h (*Schuck et al., 2008*).

## FTIR Spectral data recording and Multivariate data analysis

The infrared measurements were performed using an FTIR device (IRAffinity-1; Shimadzu) in the range of 4,600 to 400 $cm^{-1}$. The ATR unit was purged with nitrogen gas to eliminate potential interference from atmospheric $CO_2$ and moisture. Subsequently, 20 µL of each milk sample was applied directly onto the surface of the ATR crystal and analyzed against a background spectrum recorded from a clean, empty prism. Exactly 45 FTIR scans were recorded for each sample in one min to get a representative FTIR spectrum. The ATR unit was cleaned using deionized water in between measurements.

For multivariate data analysis, baseline correction was performed (Fig. S1) for removing background noise and instrument artifacts. Then the region between 4,600 and 400 $cm^{-1}$ was selected, and each spectrum was mean-centered, and standardized. FTIR data were exported to multivariate data analysis using SIMCA-P version 14.1 software package (Umetrics, Umeå, Sweden).

# RESULTS & DISCUSSION

## Water activity and moisture content

Water activity ($a_w$) and moisture content were initially determined for milk samples with $a_w$ values and found to be not significantly different ($p > 0.05$, one way ANOVA test), while the moisture content scores varied from 84% in BM2 to 89% in BM1, (Table S2). The $a_w$ reflects the water availability; the greater the $a_w$ value, the higher the water availability in milk sample, which facilitates microbial and biochemical changes in the milk. There was also a strong correlation between $a_w$ and moisture content as previously reported (*Schuck et al., 2008*).

## FTIR fingerprinting of milk samples

FTIR spectral analysis (Fig. S2) revealed similarities between milk samples in context to the main components, such as fats, proteins, polysaccharides, and water. However, differences in the intensities of the FTIR bands were observed, corresponding to varying concentrations of these components. FTIR spectral analysis revealed distinct molecular fingerprints for buffalo (BM), cow (CM), goat (GM), and camel (LM) milk, with characteristic absorption bands reflecting quantitative and qualitative differences in their fat, protein, sugar, and water content, consistent with prior studies. FTIR bands at 2,870–1,464 $cm^{-1}$ corresponded to -CH groups of the fat component detected in IR spectra of all milk samples at different intensities, which is attributed to differences in fat levels in each milk sample. Integration of FTIR spectral peaks from each milk sample further substantiated the structural identification of key milk components within the mid-infrared (mid-IR) region (*Eid, el Shamy & Farag, 2022*). Several factors, including experimental settings, preprocessing techniques, and the inherent variability in milk composition influenced by diet, breed, lactation stage, and environment may affect FTIR spectrum in addition to adulteration (*Souhassou et al., 2018*).

The ATR–FTIR spectra revealed characteristic functional groups corresponding to key milk constituents. A broad and intense absorption band observed between 3,700–3,000 cm$^{-1}$ was attributed to O–H stretching vibrations, indicative of water content (*Santos, Pereira-Filho & Rodriguez-Saona, 2013*). Sharp absorption bands in the region of 2,870–1,464 cm$^{-1}$ corresponded to C–H stretching vibrations, primarily arising from lipid components (*Nicolaou, Xu & Goodacre, 2010*). Protein content was evident from the presence of amide I and amide II vibrational bands, detected at approximately 1,600–1,548 cm$^{-1}$ (*Aernouts et al., 2011b*), along with a distinctive O = P–O stretching band near 1,100 cm$^{-1}$, attributed to casein—the predominant milk protein (*Etzion et al., 2004*). Lactose levels were reflected by absorption bands between 1,159–1,076 cm$^{-1}$, consistent with sugar-related vibrations (*Aernouts et al., 2011a*; *Al Otaibi, Bakir & Afkar, 2019*; *Nicolaou, Xu & Goodacre, 2010*; *Yaman, 2020*). Additionally, spectral features observed within the 1,200–400 cm$^{-1}$ range corresponded to various functional groups, including C–H bending, C = O stretching, and in-plane bending of C–O–H, which were associated with a range of milk constituents such as carbohydrates, lipids, amino acids, and organic acids (*Santos, Pereira-Filho & Rodriguez-Saona, 2013*).

## FTIR-based multivariate data analysis of milk samples
### Unsupervised PCA and HCA of milk samples using FTIR
The integrated FTIR dataset was subjected to unsupervised PCA and supervised OPLS-DA multivariate data analysis of the four milk types: buffalo (BM), cow (CM), goat (GM), and camel (LM) milk. PCA was used for initial sample segregation and variance exploration, while OPLS-DA enhanced discrimination and identified key markers from score, loading and contribution plots. The analysis covered the fingerprint region (1,499–400 cm$^{-1}$) and the remaining spectral regions (4,600–1,500 cm$^{-1}$).

PCA score plots (Fig. 1A) encompassed by the first two components explaining 85.9% and 8.28% of the variance, with LM and GM exhibiting distinct separation from BM and CM, which clustered more closely. LM and GM milks have generally positive scores for PC1, whereas BM and CM milks have negative scores. For the PC2 scores, milks of CM and GM had positive scores, while those of BM and LM milks had negative scores. An unsupervised PCA model (Fig. 1A) for BM, CM, GM and LM across the whole FTIR spectra (4,600–400 cm$^{-1}$) revealed a score plot where the first two components highlighted GM separation in the upper right quadrant, while LM separated in the lower right quadrant. While BM and CM remained slightly intermixed, another PCA model (Fig. 1B) was developed using the fingerprint region (1,499–400 cm$^{-1}$) and another PCA model (Fig. 1C) using the remaining spectral regions (4,600–1,500 cm$^{-1}$), but these models failed to separate BM and CM completely, indicating the need for OPLS-DA for improved resolution.

The 3D PCA plot (Fig. 1D) further confirmed this separation across four components. This prompted a separate PCA analysis with clustering focusing on BM and CM for better discrimination. HCA was used for samples clustering in an improved graphical manner and to be further compared to results derived from the GC-MS analysis of the same samples.

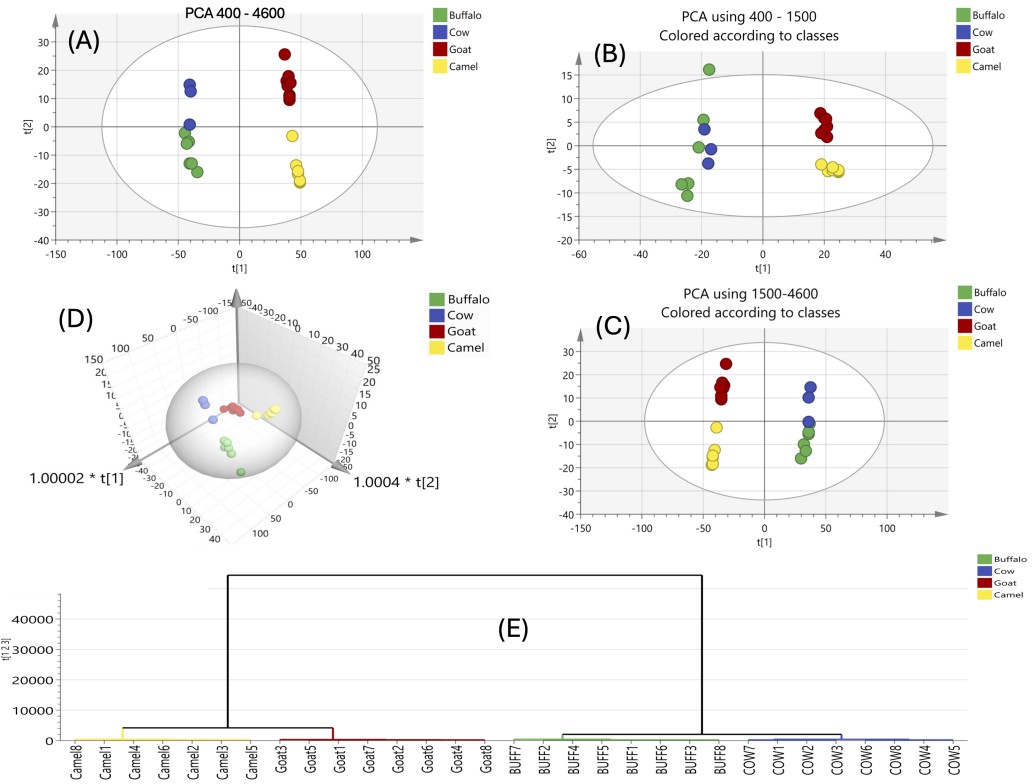

**Figure 1** **FTIR-based PCA score plot and HCA of all analyzed milk samples using FTIR for its finger-print region 1,499–400 cm$^{-1}$ and region from 4,600–1,500 cm$^{-1}$.** FTIR-based PCA score plot and HCA of all analyzed milk samples using FTIR for the whole IR region 4,600–400 cm$^{-1}$ (A), the fingerprint region 1,499–400 cm$^{-1}$ (B) and region from 4,600–1,500 cm$^{-1}$ (C). 3D scores plot of the whole FTIR region showing separated milk types. HCA of the whole IR region confirmed the ability of the model for differentiation of 4 milk types (D).

HCA was applied to the FTIR spectra of milk types using the whole region of their FTIR spectra, showing good ability of PCA model for separation of the four milk types.

### Supervised OPLS-DA of all milk samples using FTIR

The OPLS-DA model, applied to all milk samples, produced a score plot (Fig. 2A) showing clear segregation of LM and GM, with a complete separation of BM and CM confirmed by appling the OPLS-DA model to BM and CM samples only, where they were completely separated as shown in 3D scores plot (Fig. 2B).

The differentiation of buffalo (BM), cow (CM), goat (GM), and camel (LM) milk types, as revealed by FTIR-based PCA and OPLS-DA analyses, is primarily driven by distinct functional groups associated with their main components. LM appeared to be different from other milk types as shown in the OPLS score plot (Fig. 2A) and in agreement with PCA results (Fig. 1A).

Coefficient plots shown in Fig. 2C of all variables in milk types was used to visualize the importance of variables in a model by showing the regression coefficients with their corresponding confidence intervals. These plots help in understanding which variables

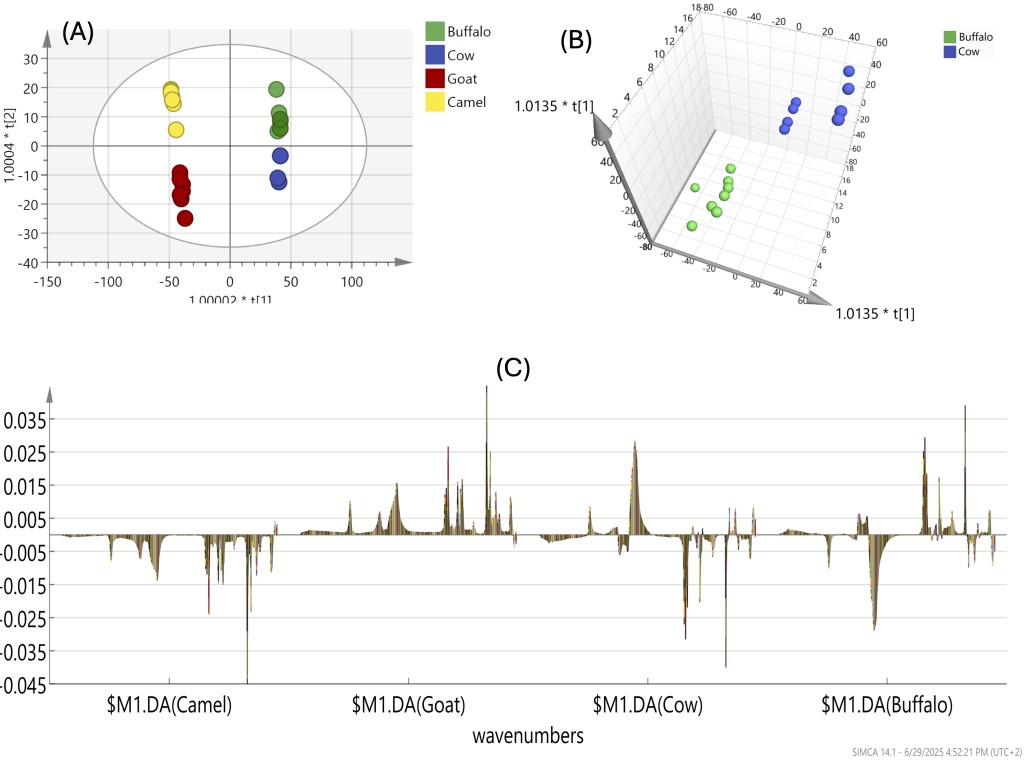

**Figure 2** (A) FTIR-based OPLS of all milk types (BM, CM, LM & GM) analyzed for the whole IR region. (B) OPLS 3D scores plot of BM and CM types showing complete separation. (C) Coefficient plots of all variables in milk types that visualize the importance of variables in a model by showing the regression coefficients with their corresponding confidence intervals.

significantly contribute to the model's predictions and can be used to identify potential outliers or influential variables. As shown in the coefficient plot (Fig. 2C), camel milk (LM) is characterized by a high fat and protein content, as indicated by FTIR-based vibrational absorption bands of the amide I (1,650–1,660 cm$^{-1}$) and amide II (1,550 cm$^{-1}$) groups, which are associated with the protein secondary structure. Additionally, the fat content is reflected by the C-H stretching vibrations of (-CH$_2$) groups, observed as sharp absorption bands in the region 2,846–2,964 cm$^{-1}$, correlating with the higher fat levels in camel milk. In contrast other milk types exhibited higher lactose content, evidenced by a sharp peak at 1,076 cm$^{-1}$, attributed to C-O stretching vibrations of lactose. Such difference in lactose content may contribute to the better flavor profile of these milk types compared to camel milk (*Sun, 2009*).

The OPLS-DA score plot (Fig. 2A) showed a good discrimination of GM from BM and CM, which were later clustered together. Such segregation was attributed to the CH$_2$ absorption band at *ca.* 2,927 cm$^{-1}$ corresponding to fatty acids' acyl chain and revealing the higher fat composition in GM compared to other milk types as evidenced by the stronger CH$_2$ stretching band. Goat milk stands out with a prominent CH2 band at 2,927 cm$^{-1}$, signifying higher fatty acid content, additional lipid-related bands at 1,747 cm$^{-1}$ (C=O

stretching), and a unique C-O-C stretching band at 1,160 cm$^{-1}$ associated with glycolipids. BM and CM were clustered together in a new OPLS model that shows complete separation as shown in Fig. 2B. Buffalo and cow milk exhibit a significant lactose-related peak at 1,076 cm$^{-1}$ (C-O stretching), correlating with higher carbohydrate content, with CM also showing a distinct N-H bending band at 1,540 cm$^{-1}$ and BM featuring a P=O stretching band at 1,240 cm$^{-1}$ indicative of phospholipids, with CM showing slightly stronger absorption, suggesting marginally higher component levels as shown in Fig. 2C. These functional groups underscore the unique protein, fat, and lactose profiles that distinguish each milk type, aligning with compositional variations reported in dairy science studies.

To evaluate the prediction performance of the FTIR-based chemometric models, a prediction set was employed. A subset of milk samples (one replicate per milk type, $n = 4$) was reserved as an independent test set, separate from the training set used to build the PCA and OPLS-DA models. The OPLS-DA models demonstrated strong predictive capability indicating robust classification of milk types in the prediction set. The classification accuracy for the test set was >95%, confirming the model's ability to generalize to new samples. This approach required no additional sample preparation or reagents, aligning with the green analytical chemistry principles emphasized in the study.

## Primary metabolites profiling of milk samples using GC-MS analysis

The six milk samples (BM1, BM2, CM1, CM2, GM, and LM) were further subjected to GC-MS analysis post-silylation for primary metabolites profiling to aid in the exact determination of low molecular weight metabolites heterogeneity in milk samples. Results revealed quantitative and qualitative differences in peaks detected in all milk samples, with a total of 87 identified peaks belonging to alcohols, amino acids/nitrogenous compounds, fatty acids, organic acids, sterols, sugars, sugar acids, sugar alcohols, and vitamins (Table S3) and (Fig. 3).

### Sugars

Among low molecular weight metabolites, sugars represented the most abundant free primary metabolite class in all milk samples. Sugars were represented by 23 peaks contributing to milk's sweet taste and flavor. The highest sugar level was detected in CM, BM, and GM whereas LM contained the least sugar level. Lactose (milk sugar) was the major sugar detected in all milk samples, with the highest lactose level accounted for CM2 representing 1.07-fold higher than that in LM which constituted the least lactose level, and in accordance with FTIR results revealing that LM is most suited for lactose intolerance (*Cardoso et al., 2010*). Variations in milk lactose levels could be attributed to the type of feed (*Zou et al., 2022*) as well as several biological and physiological factors including dairy animals' health, energy balance, and metabolism (*Costa et al., 2019*).

### Sugar alcohols

The highest sugar alcohol level was detected in LM with 2.2-fold higher than CM1. Myo-inositol was the major identified sugar alcohol. LM encompassed the highest myo-inositol level at 4-fold than that in CM1. Myo-inositol contributes to the sweet taste of LM posing

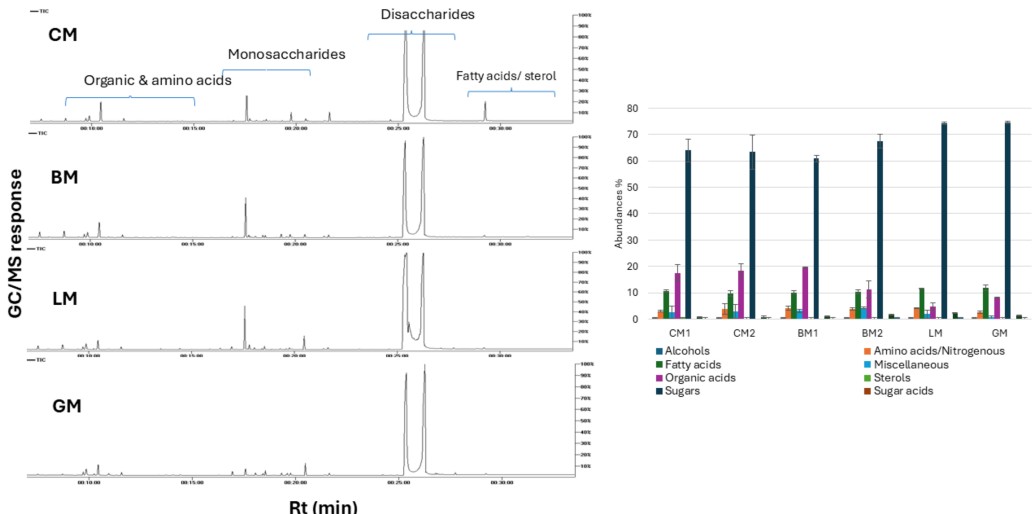

**Figure 3** Representative GC-MS chromatograms of primary metabolites in BM, CM, GM, and LM (A) and relative percentile levels in analyzed milk specimens (B).

it as a low-calorie source of milk compared to other milk types (*Li et al., 2022*; *Fukuda, 2013*).

## Sugar acids

Sugar acids constituted a minor class in all milk samples, represented by glucuronic acid derived from glycans hydrolysis through $\beta$-glucuronidase (*Sunds et al., 2021*).

## Organic acids

Organic acids comprised the second major class of primary metabolites in milk. Organic acids provide an acidic sour characteristic taste in addition to its natural preservative effect (*Zhu et al., 2020*). Organic acids also aid in protein digestion and enhance its utilization (*Farag, Mohsen & Abd El Nasser, 2018*). The highest organic acids level was detected in BM1 and CM2, whereas the lowest level was detected in LM. The organic acid level in BM1 was 5.2-fold higher than that in LM, accounting for the higher acidity (sourness) of other milk types compared to LM. Citric acid and its isomer were the major forms detected in BM and CM, compared with much lower level in LM. Citric acid level in CM2 was at 9.8-fold level than in LM. Citric acid is formed as an intermediate product in the tricarboxylic acid cycle or from the dairy animal's diet and aids in carbohydrate metabolism (*Li & Jiang, 2019*).

## Fatty acids

Free fatty acids constituted the third major class of primary metabolites in all milk types. CM1 showed the highest level of fatty acids *versus* lowest level in LM. The fatty acids content in CM1 was at 1.13- fold higher than LM. Furthermore, an unidentified fatty acid was found to be the predominant fatty acid across all milk samples, with particularly high abundance in GM, consistent with the FTIR findings (Fig. S2). This was followed by

the detection of long-chain saturated fatty acids—palmitic and stearic acids—which are typically transferred from the animal's diet to the mammary glands (*Zhang et al., 2011*). Dietary enrichment with palmitic and stearic acids has been shown to enhance milk fat yield and overall production (*Loften et al., 2014*). n-Decanoic acid (capric acid) detected in all milk samples imparts a buttery and milky odour and flavour (*Farag et al., 2020*). The medium-chained fatty acids and free fatty acids are produced *via* lipolysis of milk triglycerides (*Khattab et al., 2019*).

## Amino acids and nitrogenous compounds

Proteolysis or casein milk degradation by milk endogenous proteinases and proteolytic enzymes leads to the formation of small and intermediate peptides and free amino acids. These free amino acids serve as precursors of various aroma compounds (*Khattab et al., 2019*). Amino acids and nitrogenous compounds were detected at highest levels in BM and CM *versus* GM and LM. Valine, the major amino acid in all milk specimens, is involved in metabolic pathways for the production of branched-chain fatty acids that contribute to the flavor of dairy products (*Caboni et al., 2019*). Urea, the most abundant nitrogenous compound, was found at elevated levels in BM1 and LM; 1.5-fold higher than that in GM. An increase in proteins in dairy animals' diets led to an increase in urea content in milk, which impairs milk characteristics (*Scano et al., 2020*).

## Vitamins

LM vitamins' content was the highest compared to other milk samples. Major vitamins included water-soluble ribonic acids (folic or vitamin B9) and ascorbic acid.

## Alcohols and sterols

Alcohols constituted a minor class in milk, with glycerol as the major alcohol component in all milk samples. Glycerol is released upon hydrolysis of triglycerides by lipases (*De Moura Aguiar et al., 2019*) and likely contributes to the sweet note in milk (*Kholif, 2019*). Cholesterol was the only detected sterol in milk at trace levels in all specimens. Moreover, LM fat globules are small, which explains their rapid digestion. Meanwhile, LM plays a significant role in lowering cholesterol levels, and human intake of LM is beneficial to human health in the long term (*Bakry et al., 2021*).

## Miscellaneous

Phosphoric acid was the only inorganic acid detected in all milk samples, detected at high levels in BM and CM compared to lower levels in GM and LM. Phosphoric acid level in BM2 was at 3.7-fold compared to GM. Phosphoric acid is characterized by its tangy taste (*Hulsebus-Colvin, 2015*). The inorganic phosphorus is generated during lactose formation in the mammary gland and subsequently detected in milk (*Scano et al., 2020*).

## GC-MS based multivariate data analysis of milk samples

Although differences in low molecular weight primary metabolites composition could be revealed from simple visual inspection of GC-MS chromatograms (Fig. 3), the dataset was analysed using multivariate data analysis to assist in samples discrimination, markers identification, and to further be compared with FTIR spectra derived models (Fig. S2).

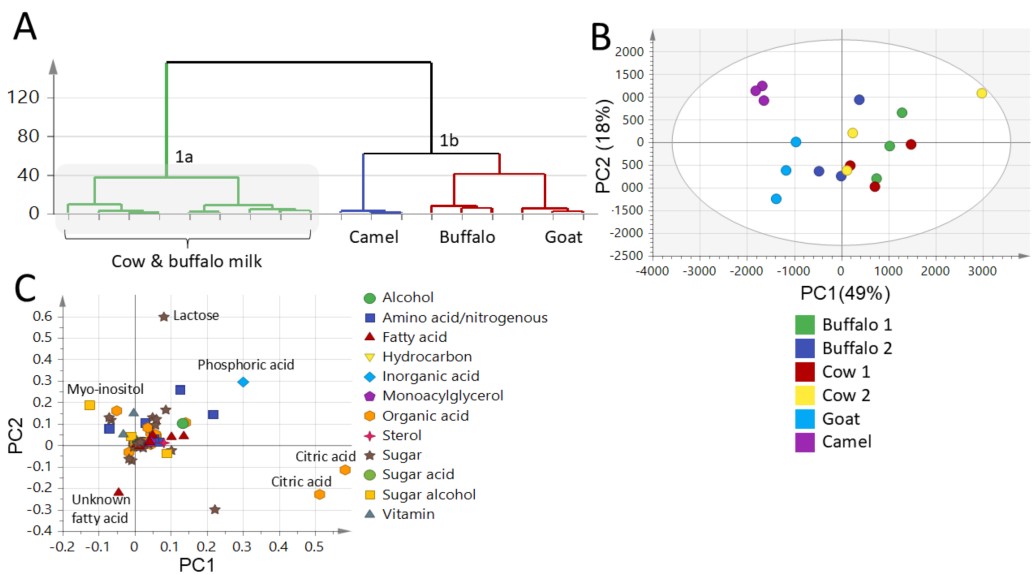

**Figure 4  GC-MS-based hierarchical clustering and principal component analysis of primary metabolites from different milk types (A) HCA plot. (B) Score plot of PC1 *vs.* PC2 scores. (C) Loading plot for PC1 & PC2 contributing metabolites and their assignments.**  The metabolome clusters are located at distinct positions in two-dimensional space described by two vectors of principal component 1 (PC1) = (49%) and PC2 = (18%).

## Unsupervised PCA and HCA data analysis of milk samples using GC-MS

PCA and HCA were applied to assess variance within milk samples in an unsupervised manner. PCA was employed for samples segregation (discrimination) and markers identification. The PCA score plot (Fig. 4B) was set by two orthogonal PCs with PC1 accounting for 49% of the total variance *versus* PC2 accounting for 18% variance. The PCA score plot succeeded in segregating milk samples into three main clusters in accordance with FTIR dataset modelling (Figs. 1 & 2). The GM and LM were well segregated, whereas BM and CM were clustered together. The markers appeared in the PCA loading plot (Fig. 4C) responsible for such segregation included myo-inositol which was found to be abundant in LM contributing to its sweet taste, in addition to the unknown fatty acid that was high in GM contributing to its goaty odour, compared to the richness of lactose, phosphoric acid and citric acid in BM and CM accounting for their sweet slightly sour taste.

## Supervised OPLS-DA of milk samples using GC-MS

Two supervised OPLS-DA models were further constructed as another attempt to identify better markers and to improve the classification potential of milk samples. The first OPLS-DA model constituted of LM *versus* CM and BM together in one class group. The OPLS-DA score plot (Fig. 5A) showed better discrimination as LM was segregated from CM and BM, which appeared to cluster together with high prediction power of $Q2 = 0.91$ (Fig. 5A), and in accordance with FTIR results. The S-loading plot identifies compounds

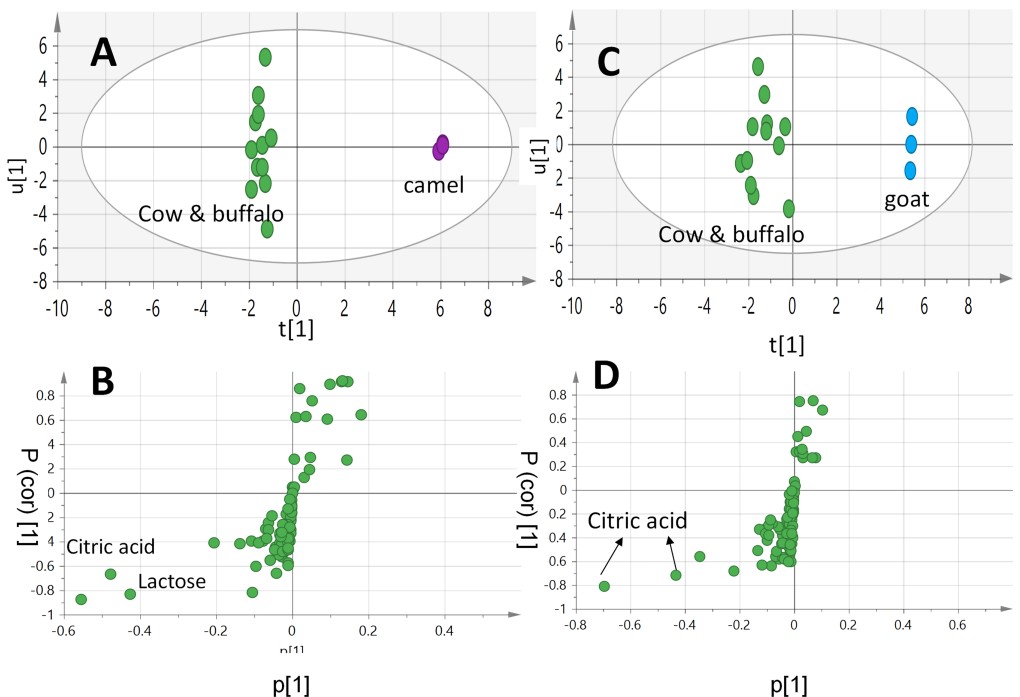

**Figure 5  GC-MS-based OPLS-DA score plot derived from modelling (A): Camel milk (LM) against both cow milk (CM) and buffalo milk (BM) in one group, (C): goat milk (GM) against both cow milk (CM) and buffalo milk (BM) in one group. Their respective S-loading plots (B) and (D) show the co-variance p[1] against the correlation p(cor)[1] of the variables of the discriminating component of the OPLS-DA model.** Cut-off values of $P < 0.05$ were used; selected variables are highlighted in the S-plot with identifications.

(markers) responsible for samples segregation. The S-loading plot (Fig. 5B) revealed that lactose and citric acid could be identified as the most discriminatory variables for both BM and CM that could contribute to BM and CM sweet slightly sour taste, though it failed to identify a clear marker for LM. Additionally, a second supervised OPLS-DA model was constructed to improve the classification potential of GM *versus* CM and BM. The OPLS-DA score plot (Fig. 5C) showed better separation of GM from clustered BM and CM specimens with prediction power of Q2 = 0.66 (Fig. 5C) confirming FTIR results. The S-loading plot (Fig. 5D) revealed that citric acid was more enriched in BM and CM contributing to their slightly sour taste, but still failed to identify a clear marker for GM. The OPLS-DA score plots of both FTIR and GC-MS datasets succeeded in the discrimination of LM and GM from each other and other milk types (BM and CM).

Lastly, a supervised OPLS model constructed to discriminate between CM *versus* BM showed values for Q2 = 0.22, and *p* value = 0.26, indicating a non-significant model due to its large *p*-value and low prediction power (Fig. S4). The FTIR-based model identified lactose and proteins as markers for BM and CM, while GC-MS identified lactose, phosphoric and citric acids as markers for BM and CM. Phosphoric and citric acids were not detected in FTIR, highlighting the benefit of complementing the two different spectroscopic

techniques in milk types' classification. Likewise, FTIR-based models identified proteins and fats as markers for LM *versus* the myo-inositol revealed when using GC-MS.

## Greenness profiles assessment

Analytical methods' greenness assessment such as the 12 GAC, AGREE, NEMI, ESA, and ComplexGAPI algorithms (*Attia et al., 2024*; *Gałuszka et al., 2012*; *Keith, Gron & Young, 2007*; *Nowak et al., 2020*; *Pena-Pereira, Wojnowski & Tobiszewski, 2020*; *Płotka-Wasylka & Wojnowski, 2021*), were employed for the comparative assessment of FTIR and GC-MS analytical methods. The analytical methods' greenness assessment aims to provide insights into the relative environmental impact and suitability for analytical applications of both the FTIR and GC-MS in terms of their safety, automation, cost, energy and solvent consumption, and wastes production. The analytical methods' greenness assessment seeks safe, fully automated, cost-effective, fewer sampling steps, and both the least waste production and solvents consumption, in addition to eliminating the use of hazardous substances. The analytical methods' greenness assessment aims to introduce a cost-effective, sensitive, eco-friendly, and economical alternative to the traditional methods known for their non-green nature and high costs, particularly in the field of quality control and food analysis.

## Challenges and opportunities according to the 12 GAC principles

The FTIR coupling to chemometric tools showed comparable performance relative to GC-MS for milk components and analysis used, as revealed from the derived score plot. GC-MS is one of the most useful techniques for the analysis of volatile and primary metabolites, and pharmaceutical ingredients, and extends herein to include milk analysis targeting its low molecular weight components for the first time. Nevertheless, from the GAC (*Gałuszka, Migaszewski & Namieśnik, 2013*) point of view, GC-MS sufferS many obstacles such as several experimental trials leading to the increase in time of analysis, waste generation, risk of exposure to hazardous gases, energy consumption, and cost (*Armenta & de la Guardia, 2016*; *Korany et al., 2017*; *Lehotay et al., 2001*). In contrast, FTIR was found to reduce such constraints being a greener vibrational spectroscopic technique that better meets the 12 principles of (GAC) (*Gałuszka, Migaszewski & Namieśnik, 2013*) as follows:

**Direct analytical technique:** FTIR is considered as a straightforward technique used for the qualitative and quantitative analysis of solid, liquid, and gas samples without pretreatment (derivatization and lyophilization). The active functional groups exhibit unique bands in the FTIR fingerprint region ($1,500–400$ cm$^{-1}$), permitting metabolites rapid identification. In the current study, milk samples were analysed by placing samples directly on the ATR unit of the FTIR device, while in case of GC-MS, milk samples were lyophilized and derivatized before analysis (additionally needing several steps). However, it is not possible to conduct metabolite identification for all metabolites by FTIR, since major functional groups are common to diverse biomolecules. Furthermore, the quantitative analysis also implies calibration process, at sensitivity and specificity usually much lower than that of GC-MS technique.

**Minimal sample size and number:** FTIR spectra provide valuable insights into the chemical structure and concentration of metabolites by identifying functional groups
through characteristic absorption bands and their relative intensities. In contrast, GC–MS is a chromatographic technique that requires extensive method development and optimization such as temperature programming, sample derivatization, and adjustment of retention gaps to achieve effective separation. These requirements can limit its throughput when analyzing large sample sets, especially compared to the more rapid and straightforward IR-based methods. As a result, multiple experimental trials and considerable sample volumes are often necessary to establish a reliable calibration model for GC–MS. In contrast, FTIR spectra were obtained rapidly using a single scan as each spectrum is considered a fingerprint. Finally, the samples size required to get representative results is reduced compared to GC-MS.

*In situ* **measurements:** The presence of handheld FTIR instruments permits the continuous measurement of samples in a cost-efficient manner. Thus, enabling the transfer from the benchtop laboratory experiments to onsite measurements, while GC-MS is a benchtop device that requires samples transfer to the laboratory.

**Integration of analytical processes:** FTIR and GC-MS integration with chemometric tools permits multiple component determination in one sample, reducing the number of samples required for full analysis and consequently minimizing energy consumption and waste generation.

**Automated and miniaturized methods:** The benchtop FTIR and GC-MS devices are equipped with an autosampler permitting the automated analysis of several milk samples on a large scale. Also, FTIR can be miniaturized by being coupled to nanotechnology in surface-enhanced infrared spectroscopy that is considered as lab on a chip device (*Eid, el Shamy & Farag, 2022*).

**No derivatization step:** The compound intermolecular chemical bonds and functional groups have unique IR vibrational frequencies depending on their structures (*Moore, 2016*). Consequently, the compound (metabolite) IR spectrum is considered a characteristic fingerprint, allowing its instant analysis without derivatization. However, the derivatization step is needed in GC-MS analysis to ensure the volatilization of primary metabolites and to increase their thermal stability to improve their identification and detection.

**Minimal analytical waste:** FTIR analysis is a straightforward technique conducted for samples without any additional solvents in either solid, liquid, or gas conditions. In the current FTIR method, only 20 µL from each milk sample was used to obtain the results. Each sample was measured for less than one minute permitting short operation time, less energy consumption, and less waste generation compared to a longer time for analysis (30 min) in the case of GC-MS.

**Multi-analyte or multi-parameter methods:** In FTIR analysis, compounds can be identified through their unique absorption patterns, particularly within the fingerprint region. When combined with chemometric tools, FTIR enables resolution of overlapping bands and allows efficient extraction of detailed chemical information from each sample. In contrast, GC–MS offers highly accurate identification and quantification of multiple metabolites; however, the development of robust methods for multi-component analysis is often time-consuming, technically complex, and associated with greater solvent usage and waste generation.

**Minimal energy usage:** In the current milk analysis, FTIR device can perform more than 45 scans for each sample in 1 min and provide the result permitting short operation time along with less energy consumption. Also, the portable FTIR instruments are battery-powered, while GC-MS instruments are laboratory benchtop device that need a longer time for the method development and conditions optimization consuming much more energy.

**Reagents obtained from a renewable source:** Helium and hydrogen are utilized in GC-MS analysis as carrier gases, most of which are expensive and subjected to availability problems. In addition, the use of derivatizing agents (pyridine) solvent is not favored due to its health hazards. In contrast, FTIR can be used for the analysis of samples without pretreatments (derivatization) in their solid, gas, or liquid condition. Moreover, many samples do not need solvents and can be measured using the FTIR devices by direct mixing with KBr for qualitative purposes as in milk samples analysis.

**Elimination of toxic reagents:** FTIR device has the advantage of samples analysis without derivatization, whether in solid, gas, or liquid state, permitting the elimination of toxic solvents. Milk samples were placed directly on the ATR cell and measured without pretreatment. However, in the GC-MS analysis, the derivatization step and its toxic reagents (pyridine) could not be eliminated.

**Operator safety:** FTIR analysis is a safe analytical method with advantages for the operators (analysts) in terms of rapid analysis, multicomponent determination in a single run, small volumes of samples, minimal solvents, and minimal waste generation, although the mobile phase (carrier gas) in GC-MS analysis requires special handling along with samples derivatization as in primary metabolites profiling. Most of these gases and solvents as pyridine are health-hazardous volatiles, obligating the analysts to wear face masks with filters.

While LC-MS is indeed a powerful technique for analyzing polar and semi-polar compounds and avoids the need for derivatization, the availability of GC-MS in our laboratory, along with its simpler sample preparation and less stringent operational requirements made it the preferred choice for this large scale metabolomic study.

## Greenness evaluation according to AGREE

AGREE is a simple and flexible assessment approach concerned with the 12 principles of GAC. It expresses the 12 principles of GAC through numerical weights, each is given a score (0–1), the final score is obtained by calculating the average of all the 12 principles scores, which also could be performed using Analytical GREEnness calculator (free software) (*Pena-Pereira, Wojnowski & Tobiszewski, 2020*). The results are represented as clock-like coloured pictogram. Results range from zero (dusky red) to one (dusky green), and results that are closer to 1 are greener (better). The results in Table 1 indicate that the FTIR method is greener than GC-MS, as the FTIR attains 0.89 while the GC-MS attains 0.46.

## Greenness evaluation according to NEMI

NEMI is one of the earliest methods used for greenness evaluation concerned with four different perspectives (*Keith, Gron & Young, 2007*). It is represented in the form of a circle

**Table 1  Comparative study between the FTIR and GC-MS methods based on AGREE, NEMI, ESA, ComplexGAPI, and RGB 12 algorithms.**

| Parameters | FTIR method | GC-MS |
|---|---|---|
| Application | Buffalo, Cow, Goat, and Camel milk | Buffalo, Cow, Goat, and Camel milk |
| AGREE metric[a] |  |  |
| NEMI tool |  |  |
| ESA Score[b] | 98 | 60 |
| ComplexGAPI tool[c] |  |  |
**Table 1** (*continued*)

| Parameters | FTIR method | GC-MS |
|---|---|---|
| RGB 12Algorithm[d] | 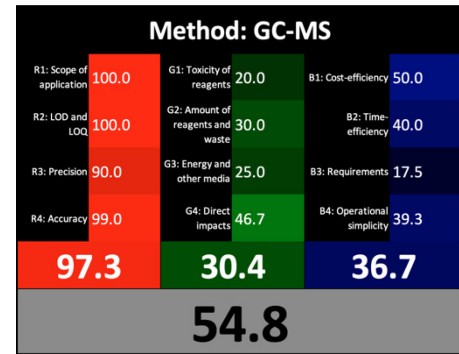 | |

**Notes.**

[a]AGREE Assessment evaluated by using the Analytical GREEnness metric approach and software.

[b]Penalty points calculation according to the analytical Eco-Scale score.

[c]ComplexGAPI Assessment evaluated by using the Analytical ComplexGAPI metric approach and software.

[d]RGB12 whiteness Assessment evaluated by using the Analytical RGB 12 algorithm.

divided into four quadrants (fields), labelled as hazardous, corrosive, waste, and PBT, each field is coloured green on fulfilling these requirements. The first field requirements ensure that the chemicals used are not classified as PBT, EPA's TRI Agency. The requirements of the second field ensures that the chemicals used in the analytical method are not considered as hazardous chemicals according to RCRA's U, P, F, D or TRI. The third field is highlighted green if the method's pH is not corrosive. The fourth field is marked green if the waste generated is less than 50 g. The FTIR method pictogram showed four green quadrants indicating a greener method, compared to GC-MS which showed only one unfulfilled quadrant attributed to the use of hazardous solvents during method development (Table 1).

## Greenness evaluation according to ESA

ESA is a semiquantitative tool for the assessment of method greenness (*Gałuszka et al., 2012*) based on deducing penalty points (analytical method parameters) that do not align with the 12 GAC principles from a score of 100. The FTIR was found to be greener (98) than GC-MS (60) as evidenced by ESA (Table 1).

## Greenness evaluation according to ComplexGAPI

Recently, ComplexGAPI has gained trust and recognition as a semi-quantitative tool. ComplexGAPI stands out for its simplicity and enhancement of the existing GAPI metric. ComplexGAPI integrates an additional hexagonal field into the original GAPI graph, following the CHEM21 parameters, representing the steps (procedures) of the analytical method (*Płotka-Wasylka & Wojnowski, 2021*). Therefore, ComplexGAPI parameters cover all the steps of the analytical method, including sampling, sample preparation, analysis, preservation, transportation, and storage. The FTIR method demonstrated superior greenness compared to GC-MS, as evidenced by greener pictograms and low E-factor. This indicates minimal waste generation, more positive environmental impact, and better sustainability (Table 1).
## Whiteness profiles assessment

*Nowak et al. (2020)* introduced a straightforward quantitative method known as the Red Green Blue (RGB) 12 algorithm for assessing the analytical method whiteness (WAC). This tool simplifies the evaluation of the analytical methods' whiteness according to the 12 WAC considerations, providing a clear assessment of sustainability. The RGB 12 algorithms are arranged into three main groups: red, green, and blue, with each group containing four algorithms. Specifically, the first group is the green group (G1-G4) concerned with the critical GAC parameters, such as toxicity levels, reagent and waste quantities, energy consumption, and direct impacts on human health, animal welfare, and genetic modifications. The second group, known as the red group (R1-R4), is concerned with the validation parameters such as the scope of application, limits of detection (LOD) and quantification (LOQ), accuracy, and precision. The third group is the blue group (B1-B4), concerned with aspects related to cost-effectiveness, time efficiency, and practical or economic feasibility. The whiteness assessment indicates the degree of adherence to the WAC principles, which is determined by aggregating the scores across all three areas (colours) using the RGB algorithm. The FTIR method showed exceptional whiteness, achieving 97.1, whereas the GC-MS method scored 54.8 (Table 1). These scores indicated the superiority of the FTIR method over GC-MS not only in terms of greenness, whiteness, sustainability, and analytical effectiveness but also in terms of economic and practical feasibility.

Finally, this study introduced a novel approach to milk analysis by integrating Fourier-transform infrared spectroscopy (FTIR) and gas chromatography-mass spectrometry (GC-MS) exemplified by buffalo (BM), cow (CM), goat (GM), and camel (LM) milk, offering a more holistic analysis than single-method studies prevalent in the literature. By leveraging FTIR's rapid, non-destructive fingerprinting of functional groups (*e.g.*, fats, proteins, lactose) and GC-MS's high-resolution detection of 87 primary metabolites (*e.g.*, myo-inositol, citric acid, phosphoric acid), coupled with chemometric tools (PCA, OPLS-DA), we achieved robust discrimination of milk types and identified unique markers, such as myo-inositol for LM and lactose for BM/CM. A key contribution lies in the systematic greenness and whiteness assessment using five metrics (12 GAC principles, AGREE, NEMI, ESA, ComplexGAPI) and the RGB 12 algorithm, demonstrating FTIR's superior sustainability (*e.g.*, AGREE score: 0.89 *vs.* 0.46 for GC-MS) and its potential as an eco-friendly alternative for routine quality control, particularly in resource-constrained settings—a dimension rarely explored in prior milk analysis studies. Additionally, our focus on camel milk, which revealed its high myo-inositol content and low lactose levels *via* GC-MS, complements FTIR's findings of elevated fat and protein content, supporting LM's suitability for lactose-intolerant individuals and its therapeutic potential for conditions like autoimmune diseases and diabetes. By addressing practical challenges in milk authenticity and quality control, such as detecting mislabeled milk types, this dual approach combines FTIR's high-throughput screening capabilities with GC-MS's in-depth research applications, offering a sustainable and practical framework for dairy industries that advances both food safety and nutritional research.

## CONCLUSIONS

This study presents a multiplex metabolomics-based approach using FTIR and GC-MS fingerprinting platforms (coupled to chemometric tools) for the characterization of metabolites in different types of milk (BM, CM, GM, and LM). Secondly, a side-by-side comparative assessment of the performance of both the FTIR and GC-MS methods was implemented in the light of green analytical chemistry principle (GAC) and white analytical chemistry principles (WAC). Both FTIR and GC-MS-based multivariate data analysis succeeded in discriminating LM and GM from other milk types, although they failed to distinguish between BM and CM, confirming their similar chemical composition. The FTIR-based analysis confirmed proteins and fats as major discriminators of LM, while GC-MS revealed myo-inositol, a sugar alcohol, as the main discriminator in LM. The FTIR-based analysis revealed that lactose and protein were the major components in BM and CM, while the GC-MS-based model revealed markers such as lactose, citric, and phosphoric acid as major components in BM and CM. Both citric and phosphoric acids were not detected using IR warranting this complementary approach in milk analysis. LM is considered one of the potential alternatives to feed infants, adults, and people allergic to CM. Our results that LM contains the least lactose level presents further added value for people suffering from lactose intolerance. In conclusion, both FTIR and GC-MS-based multivariate data analysis models identified lactose, proteins, citric and phosphoric acids as markers for BM and CM, while proteins, fats, and myo-inositol were identified as markers for LM. We do admit the limitation of the sample size, which is crucial for more robust conclusions, particularly to minimize the risk of bias due to the factors affecting milk components such as the dairy animal's physiological condition, animal feed type, biological, and enzymatic changes in milk, and the environment surrounding the milking area. This study serves as an initial exploration utilizing a comparative metabolomic approach, and future research will involve a larger and more diverse sample from other origins or to assess other variables related to milk quality.

### Funding

This article is based upon work supported by the Science, Technology & Innovation Funding Authority (STDF), Egypt under grant number 47051. This research is part of a project (Oli4food) that has received funding from the PRIMA Programme supported by the European Union's Horizon 2020 Research and Innovation Programme, project ID No. 1854. The funders had no role in study design, data collection and analysis, decision to publish, or preparation of the manuscript.

### Grant Disclosures

The following grant information was disclosed by the authors:
Science, Technology & Innovation Funding Authority (STDF): 47051.
European Union's Horizon 2020 Research and Innovation Programme: 1854.
## Competing Interests

Mohamed A. Farag is an Academic Editor for PeerJ.

## Author Contributions

- Mohamed A. Farag conceived and designed the experiments, performed the experiments, analyzed the data, prepared figures and/or tables, authored or reviewed drafts of the article, and approved the final draft.
- Sherif M. Eid conceived and designed the experiments, performed the experiments, analyzed the data, prepared figures and/or tables, authored or reviewed drafts of the article, and approved the final draft.
- Sherine El-Shamy conceived and designed the experiments, analyzed the data, prepared figures and/or tables, authored or reviewed drafts of the article, and approved the final draft.

## Data Availability

Raw data is available in the Supplemental Files.

## Supplemental Information

Supplemental information for this article can be found online at http://dx.doi.org/10.7717/peerj.19921#supplemental-information.

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
