# Peer review of "Gas chromatography-mass spectrometry and Fourier-transform infrared spectroscopy coupled to chemometrics for metabolome analysis of different milk types in the light of green analytical chemistry"

_PeerJ, doi:10.7717/peerj.19921_

## Round 0.1 · original submission · Major Revisions

Your manuscript needs careful revision before being resubmitted for evaluation. Please pay special attention to reviewer #1's comments. Please submit your revised version as soon as possible.

**Language Note:** The review process has identified that the English language must be improved. PeerJ can provide language editing services - please contact us at [email protected] for pricing (be sure to provide your manuscript number and title). Alternatively, you should make your own arrangements to improve the language quality and provide details in your response letter. – PeerJ Staff

·

Basic reporting

This manuscript reports the use of a metabolomic approach using FTIR and GC-MS to distinguish different types of milk – cow, buffalo, goat, and camel. The manuscript is well-written and the experimental design used by the authors is clear and concise. However, the study lacks in some points that are of great importance to consider it acceptable for publication.
I have one main concern related to the number of samples used by the authors do calibrate the multivariate method. And the authors also do that, as claimed in the conclusion: “We do admit the limitation of the sample size, which is crucial for more robust conclusions…”. The authors mention that the samples are prone to the variability of its main contents, such as water, sugars, fatty acids, among others. Thus, the authors use only 6 samples to perform multivariate methods to segregate the different types of milk. As can be seen in Fig. S2, the milk samples share the same profile, with slight variations due to the different concentration of some components. So, I ask: is the number of samples great enough to differentiate the samples? What if a different sample were analyzed by the FTIR-model provided by the authors, would it be possible to determine what kind of milk it is? Based on the PCA graphic the authors provided (Fig 1.), it was possible to only differentiate camel milk (LM) from the others. What about the papers the authors cite in the introduction for multivariate calibration? How many samples did they use to calibrate their methods?
With this point in mind, what is the difference between the study presented by the authors and others available in the literature to differentiate milk? All the points observed by the authors using FTIR and GC-MS are no novelty.
Also, the title of the paper highlights the green analytical chemistry (GAC). However, the authors did not propose any novelty from the GAC point of view but only evaluated a lot of metrics to show an obvious conclusion that a FTIR method is easier, faster, low-cost, etc… than a GC-MS method.
General Comments:
I strongly suggest the authors to use mass spectrometry instead of spectroscopy throughout the manuscript.
Lines 100 to 105: I suggest the authors to move these sentences to the end of the introduction, because it justifies the work developed.
Line 163: each instead of Each
Line 172: the reference for the chromatographic method should be provided at the beginning of the method description (The chromatographic method was adapted from El-Shabasy et al. 2024) or in the end. Please, choose one of these options.
Line 209-210: …baseline correction was performed (Fig. S1) the region between 4600 and 400 cm-1 was selected… It seems like something is missing in this sentence. Please, check it out.
Line 222: the information of each IR band should not be provided between parenthesis. I suggest the authors to rewrite this description.
Line 300 and beyond: The authors claim that sugars represent the most abundant free primary metabolite class in milk samples, with composition ranging from 61-75%. However, this percentage is related to the chromatographic area, which is not directly related to the concentration when comparing different classes of compounds. The chromatographic response is related to the ability of a compound in producing ions in the ion source. The higher the population of ion generated for the same concentration; the higher will be the response. It is probably what is happening for sugars. It is so true that, for example, in the way it is described by the authors, it seems like there is more sugar than water in milk, for example. In this way, my opinion is that the authors should use the results of GC-MS as they used in lines 319-322, for example, comparing the responses of each compound in the different samples (Myo-inositol was the major identified sugar alcohol. LM encompassed the highest myo-inositol level at 4-fold than that in CM1). Thus, I strongly suggest the authors to rewrite the discussion taking this information into account, not only for sugars but for all the other compounds.
Lines 422 and beyond: Greenness profile assessment
This topic is extremely relevant nowadays, specially when talking about the development of analytical methods that fit the sustainability principles. However, the way the authors discuss all the evaluated metrics seems boring. The authors discussed all the five softwares used in this study and all the 12 principles of GAC to get the obvious conclusion when comparing a FTIR and a GC-MS method. It is clear that a FTIR method seems greener than a GC-MS method. However, the response obtained when using a GC-MS is much more complete than a FTIR method will provide. Thus, my suggestion is that the authors should be more direct and select only one software, such as AGREE or any other of your choice, and make the discussion in a more direct way.

Experimental design

no comment

Validity of the findings

no comment

Additional comments

no comment

Reviewer 2 ·

Basic reporting

a) I recommend a comprehensive revision of the manuscript to improve the technical language and ensure better clarity in the methodology, data presentation, and results interpretation.
b) In the Abstract, I suggest including numerical results and specific findings to offer a more informative summary. In the Methodology section, although the focus on green analytical chemistry is commendable, it would be important to also provide detailed descriptions of the FTIR and GC-MS techniques used, along with the applied and evaluated parameters.
c) The theoretical background is appropriate and well-supported.
d) The manuscript is largely aligned with PeerJ’s structural standards. However, some figures and graphs require improvement in image resolution and consistency in formatting (font type and size).

Experimental design

a) The raw data provided are satisfactory and meet the journal’s requirements.
b) The manuscript presents original primary research that fits within the scope of PeerJ.
c) The study is relevant and significant. However, I recommend giving greater priority emphasis to the study in green analytical chemistry by applying both analytical techniques, rather than focusing the discussion mainly on the comparison of FTIR and GC-MS capabilities without extensive methodological development. The complementary use of both techniques seems more appropriate and aligned with the study objectives.
d) Although methodological descriptions are generally sufficient for replication, I recommend improving the clarity of procedures

Validity of the findings

All underlying data are provided, robust, and statistically well-analyzed.

Additional comments

Suggestions are listed below:
Introduction:
- Line 116: I suggest rephrasing the sentence for clarity. For example: “GC-MS is considered a versatile and essential technique for the qualitative and quantitative analysis of primary metabolites, with increased sensitivity for detecting compounds at lower concentrations.”
Materials and Methods:
- Lines 154–155: Specify the sterilization method and equipment used for the test tubes (e.g., autoclave, 2% NaOCl solution, or gamma radiation).
- Lines 171–174: Improve clarity in the description of GC-MS parameters. Suggested rewording: “The injector temperature was set to 220 °C. The column oven was held at 38 °C for 3 min (El-Shabasy et al., 2024). The temperature was then increased at a rate of 12 °C/min to 180 °C, held for 5 min, and then increased at 40 °C/min to 220 °C, held for 2 min. Helium was used as the carrier gas at a flow rate of 1 mL/min.” - Lines 199–200: Suggested rewording: “The moisture content (MC) of the milk samples was determined gravimetrically. The samples were transferred to an air oven (Thermo Electron Corporation, Waltham, MA, USA) and dried at 100 °C to constant weight for 2 h (Schuck et al., 2008).” - Lines 223–226: Suggested rewording: “FTIR spectral evaluation (Fig. S2) revealed similarities between milk samples in the profiles of the main components, such as fats, proteins, polysaccharides, and water. However, differences in the intensities of the FTIR bands were observed, corresponding to varying concentrations of these components.”
- Lines 226–244: Strengthen the interpretation of the results, relating them more clearly to the findings reported by the cited authors.
Figures:
- Improve the resolution of the following figures for better visualization:
- Figure_1.pdf
- Figure_2.pdf
- Figure_3.pdf (chromatograms are difficult to interpret due to large differences in peak heights; higher resolution is needed for better visualization of the baseline)
- Supplementary Figure_S1.pdf
Additional comments:
- Lines 269–270: The term “significantly different” should be statistically corroborated.
- Review figure titles for formatting, spacing, and excess text, and be more direct.
- Lines 303 to 307:
- The chromatogram in Figure 3 is difficult to interpret and needs better resolution for clarity when comparing it to the information listed in the text.


Final Remarks to the Authors:
I congratulate the authors on the relevance of the research and the robust integration of green analytical chemistry with chemometric analysis. The study demonstrates an important effort to apply analytical methods that reduce toxicity, reagent consumption, procedural steps, high costs and analysis time.
That said, I believe the manuscript would benefit from revisions in the following areas:
- Technical writing (some terms cited are not technical in the field of analytical chemistry) and English grammar (some sentences are inverted).
- Improvements in the quality of the resolution of graphs and chromatograms.
- Clearer alignment of the study objectives with the conclusions — although the volume of data is significant and supported by chemometrics, the main objective (application of green analytical chemistry via FTIR and GC-MS for analysis of milk types) is broad.

Reviewer 3 ·

Basic reporting

Dear authors,

The paper “Gas Chromatography-Mass Spectroscopy and Fourier-transform infrared spectroscopy coupled to chemometrics for metabolome analysis of different milk types in the light of green analytical chemistry” is very well written in general. The authors present a good introduction, the results are well presented and the figures are very well designed and clear.

In my opinion, after completing this data, the paper is suitable for publication in Peer J.

Experimental design

I believe that the authors failed to present a comparison of the proposed methods compared to the methods already used for the analysis of these compounds. In general, do they present advantages?

I suggest that the results presented for each of the compounds should be discussed with data already found in the literature: are the data obtained relevant results, are the data obtained in agreement with the results already published, are the data obtained different?...

Was the FTIR method developed validated? Which validation parameters were evaluated?

Validity of the findings

Very good.

---

## Round 0.2 · accepted · Accept

Dear Dr Farag,

I'm pleased to inform you that your manuscript, Gas Chromatography-Mass Spectrometry and Fourier-transform infrared spectroscopy coupled to chemometrics for metabolome analysis of different milk types in the light of green analytical chemistry, has been accepted for publication after careful consideration of the revised version. The changes made in response to the reviewers' comments have satisfactorily addressed the concerns raised.

We appreciate your efforts in improving the manuscript and look forward to its publication. Please find any final editorial steps outlined below (if applicable).

Congratulations on your achievement!

Best regards,
Eder.

·

Basic reporting

This reviewed manuscript reports the use of a metabolomic approach using FTIR and GC-MS to distinguish different types of milk – cow, buffalo, goat, and camel. The manuscript was already well-written and the experimental design used by the authors is clear and concise. As detailed in my comments, some adjustments were necessary to improve the quality of the study conducted by the authors and I’m glad they answered it. Thus, I can see a much better manuscript in the current version.
The authors took care of answering my main concern related to the number of samples used to calibrate the multivariate method. The addition of this experiment has clearly improved the quality of the results obtained by the authors, as can be seen in Fig. 1. The text added by the authors in Lines 651-670 summarizes the advantages of the study conducted by the authors. Then, I would suggest the authors to remove the statement: “We do admit the limitation of the sample size”. By the PCA plots presented it does not seem so relevant right now.
Also, the authors have improved the manuscript by adjusting some sentences and comments provided by the others reviewers. The Figures provided are much better in this current version of the manuscript.
It is my point of view that the manuscript is now acceptable for publication in PeerJ.

Experimental design

no comment

Validity of the findings

no comment

Additional comments

no comment

Reviewer 2 ·

Basic reporting

After reviewing the revised version of the manuscript, I confirm that the authors have adequately addressed the comments and suggestions provided in the previous review stage. The current version of the manuscript demonstrates significant improvements in the following aspects:
• The manuscript now employs professional, clear, and unambiguous English throughout, enhancing readability and academic tone.
• A sufficient historical and contextual background has been provided, supported by relevant literature references that frame the study within the field.
• The manuscript presents a professional structure, with well-organized sections, clear and informative figures and tables, properly formatted. Additionally, raw data have been shared, contributing to transparency and reproducibility.
• The work is self-contained, with relevant results clearly presented and aligned with the stated hypotheses and research questions.
Given the satisfactory revisions and improved quality, I consider the manuscript suitable for publication, pending the editor’s final decision.

Experimental design

After reviewing the revised manuscript, I confirm that the authors have satisfactorily addressed the comments and suggestions related to the following key aspects:
• The study constitutes original primary research that aligns well with the objectives and scope of the journal.
• The research question is clearly defined, relevant, and significant, with a clear statement on how the study fills an identified knowledge gap.
• The investigation was conducted with rigor, adhering to high technical and ethical standards.
• The methods are described in sufficient detail and clarity to allow for replication by other researchers.
Based on these improvements, I consider the manuscript to meet the scientific and editorial standards required for publication, subject to the editor’s final decision.

Validity of the findings

After reviewing the revised manuscript, I note that the authors have adequately addressed the comments and suggestions regarding the following aspects:
• The manuscript emphasizes the importance of meaningful replication, clearly and appropriately outlining the rationale and benefits to the literature;
• All underlying data have been provided, being robust, statistically sound, and properly controlled;
• The conclusions are well formulated, directly linked to the original research question, and appropriately limited to the supporting results.
I consider the manuscript to meet the scientific and editorial standards necessary for publication, pending the editor’s final decision.

Additional comments

Dear Authors,
I have reviewed the revised version of the manuscript and noted that the majority of the comments and corrections raised in the previous review have been adequately addressed. The implemented changes have contributed to improving the manuscript both in terms of clarity and technical-scientific content.
I highlight significant improvements in the following aspects:
• Clarity and textual organization: the manuscript is now more cohesive and presents a better-structured scientific argument;
• Methodological adequacy: the additional clarifications provided in the Materials and Methods section enhance the reproducibility of the procedures;
• Presentation of results: appropriate adjustments in the description and interpretation of data, with greater objectivity and support from relevant references;
• Discussion and conclusions: expanded critical analysis and better alignment of conclusions with the presented evidence;
• Formal corrections: previously noted grammatical, orthographic, and formatting issues have been resolved.
Therefore, I consider the manuscript suitable for publication, provided it meets the remaining editorial criteria of the journal.

Reviewer 3 ·

Basic reporting

No comment.

Experimental design

No comment.

Validity of the findings

No comment.

Additional comments

No comment.